# The Role of Radiomics and AI Technologies in the Segmentation, Detection, and Management of Hepatocellular Carcinoma

**DOI:** 10.3390/cancers14246123

**Published:** 2022-12-12

**Authors:** Dalia Fahmy, Ahmed Alksas, Ahmed Elnakib, Ali Mahmoud, Heba Kandil, Ashraf Khalil, Mohammed Ghazal, Eric van Bogaert, Sohail Contractor, Ayman El-Baz

**Affiliations:** 1Diagnostic Radiology Department, Mansoura University Hospital, Mansoura 35516, Egypt; 2Bioengineering Department, University of Louisville, Louisville, KY 40292, USA; 3Faculty of Computer Sciences and Information, Mansoura University, Mansoura 35516, Egypt; 4College of Technological Innovation, Zayed University, Abu Dhabi 4783, United Arab Emirates; 5Electrical, Computer, and Biomedical Engineering Department, Abu Dhabi University, Abu Dhabi 59911, United Arab Emirates; 6Department of Radiology, University of Louisville, Louisville, KY 40202, USA

**Keywords:** deep learning, machine learning, AI, computed tomography, hepatocellular carcinoma

## Abstract

**Simple Summary:**

As a primary hepatic tumor, hepatocellular carcinoma (HCC) is the most prevalent kind. Recent developments in magnetic resonance imaging (MRI) and computed tomography (CT) have the potential to enhance detection, segmentation, discrimination from HCC mimics, and tracking of therapy response. Radiomics, artificial intelligence (AI), and related methods have already been used with promising results in other diagnostic imaging fields. In this review, we briefly cover the radiomics and AI clinical applications for the identification, segmentation, and management of HCC. In addition, we also look into their potential to improve HCC diagnosis and provide appropriate treatment planning. Finally, overall limitations and future directions of the research in this field are outlined.

**Abstract:**

Hepatocellular carcinoma (HCC) is the most common primary hepatic neoplasm. Thanks to recent advances in computed tomography (CT) and magnetic resonance imaging (MRI), there is potential to improve detection, segmentation, discrimination from HCC mimics, and monitoring of therapeutic response. Radiomics, artificial intelligence (AI), and derived tools have already been applied in other areas of diagnostic imaging with promising results. In this review, we briefly discuss the current clinical applications of radiomics and AI in the detection, segmentation, and management of HCC. Moreover, we investigate their potential to reach a more accurate diagnosis of HCC and to guide proper treatment planning.

## 1. Introduction

The most prevalent hepatic malignant tumor in the world is hepatocellular carcinoma (HCC). It has a high morbidity and mortality rate, being the fourth most common cause of cancer death [1]. Predisposing factors include chronic liver disease, cirrhosis, prior infection with hepatitis B or C virus, and ingestion of aflatoxins [2]. Initial diagnosis of such neoplasms and further monitoring of therapeutic response is achieved through several imaging modalities, along with serum biomarkers such as alpha-fetoprotein (AFP). In the majority of cases, triphasic computed tomography (CT) or dynamic magnetic resonance imaging (MRI) provides an accurate diagnosis, and pathological analysis is only carried out in the minority of patients in which imaging features are ambiguous or not specific. In recent years, artificial intelligence (AI) has become involved in almost all diagnostic and prognostic programs, to increase accuracy, eliminate human errors, and save time and money. Current AI models are trained on large amounts of derived imaging data, known as radiomics. Such data are usually acquired in five steps: (a) image acquisition, (b) segmentation, (c) feature extraction, (d) exploratory analysis, and (e) modeling [3]. Therefore, radiomics is simply the use of computer-based algorithms to extract quantitative features from a conventional image, followed by a correlation of these features with pathology or clinical response [4].

Although the terms AI and machine learning (ML) are used interchangeably in the area of imaging-based medical systems, AI is defined as a broad concept of how a computer methodology can imitate human intelligence. ML is a subset of AI in which computer models are trained on large sets of data and then adapt as new data is presented. Digitization of radiological images is now the medical standard in developed countries, and imaging data are stored in picture archives and communication systems (PACS). The clinical image interpretations are stored in radiology information systems. Additionally, AI advancements in the area of image processing, increased computer processing capacity, low-cost data storage, and fast data transfer rates have played an important role in the widespread of use AI-based applications in addressing various radiology-based challenges. This has led to an increased number of publications using AI for medical image analysis, including disease identification, segmentation, classification, and outcome prediction. Because the data archived in radiology departments is commonly unstructured and patient-centered, the data conglomerate cannot be used for training AI, and implementing AI projects requires a pipeline consisting of multiple steps which begin with the identification and extraction of relevant imaging data and clinical reports. The main role of the radiologist with regard to commercial AI products is to be an informed user of these tools. In order to be an informed customer, radiologists must understand the clinical need for AI tools in particular clinical use cases, conduct systematic evaluations of AI tools prior to implementing them in practice, and apply clinical experience while avoiding the pitfalls of over-reliance on technology.

In this article, we present a thorough review of research spanning the last two decades on AI- and ML-based methods in the management of HCC. The reviewed data were available from different search engines, including ResearchGate, PubMed, and Google Scholar. In the twenty-year range from 2002 to 2022, we focused on manuscripts primarily concerned with HCC. In our search, we used different combinations of the following terms: hepatocellular carcinoma, HCC, deep learning, machine learning, artificial intelligence, computed tomography, magnetic resonance, detection, segmentation, management, treatment response, radiomics, etc., resulting in a total of >130 studies that met our inclusion criteria. Studies that utilized AI or ML and satisfied the aforementioned inclusion criteria received higher priority in the search. To the best of our knowledge, no reliable AI-based algorithms exist that can be used as a standalone gold standard for segmentation, detection, or management of HCC. By analyzing the included studies, we intend to pave the way for future research into the development of precise AI-based diagnosis and prediction systems for the best possible personalized HCC treatment.

The next section presents a background on the different techniques of AI and radiomics models. Section 3 reviews studies that combine different markers with AI and ML to build computer-aided models for hepatic segmentation. Section 4 outlines the different techniques used for differentiation of HCC from other hepatic observations. Section 5 reviews the different AI techniques for management of HCC. Section 6 summarizes the limitations of this study and describes future perspectives. Finally, Section 7 concludes the paper. A summary of the different applications of AI and radiomics in the field of HCC is illustrated in Figure 1.

## 2. Background on AI and Radiomics Techniques

### 2.1. Artificial Intelligence (AI)

The term “artificial intelligence (AI)” refers to the computational capacity to carry out tasks that are comparable to those carried out by humans, with varying degrees of autonomy, to process unique raw data (inputs) and produce useful information (outputs) [5]. While the foundations of AI were laid decades ago, it was not until the advent of modern powerful computational technology, coupled with the ability to capture and store massive data quantities, that it became possible to realize AI capabilities in the tasks most important to radiology, such as pattern recognition/identification, preparation, object/sound recognition, problem-solving, disease prognostication, assessing the necessity of therapy, and providing patients and clinicians with prognostic data on treatment outcomes. Despite the fact that healthcare is a daunting area in which to apply AI, medical image analysis is becoming a very important application of this technology [6]. From the outset, it has been obvious that radiologists could benefit from the powerful capabilities of computers to augment the standard procedures used for disease detection and diagnosis. The use of computer-aided diagnosis (CAD) systems, which were forerunners to modern AI, has been encouraged to help radiologists detect and analyze potential lesions in order to distinguish between diseases, reduce errors, and increase efficiency [7]. As the nature of CAD systems is that they are tailored to a specific task, their variable reliability and the possibility of false positive results require that a qualified radiologist confirm CAD findings [7]. Consequently, continuous efforts are being made to improve the efficiency and accuracy of CAD and to promote the assistance that it can offer in routine clinical practice. The development of artificial neural networks (ANN) in the middle of the 20th century [8] and their subsequent progression, which has brought forth the concepts of machine learning (ML), deep learning (DL), and computational learning models, are the primary reasons for the rise of AI.

Machine learning (ML) is one of the most important applications of AI. The training process, in which computer systems adapt to input data during a training cycle [9], is the cornerstone of ML. Such models require large amounts of high-quality input data for training. The creation and use of large datasets structured in such a way that they can be fed into an ML model are sometimes referred to as “big data”. Through repeated training cycles, ML models can adapt and improve their accuracy in predicting correct data labels. When an appropriate level of accuracy is achieved, the model can be applied to new cases which were not a part of the training stages [10,11]. ML algorithms can be either supervised or unsupervised, depending on whether the input data are labeled by human experts or unlabeled and directly categorized by various computational methods [10,12]. An optimal ML model should include both the most important features needed to generate desired outputs and the most generic features that can be generalized to the general population, even though these features may not be defined in advance. Pattern recognition and image segmentation, in which different meaningful regions of a digital image (i.e., pixels or segments) are identified, are two common ML tasks in radiology. Both have been successfully used for a variety of clinical settings, diseases, and modalities [13,14,15].

Deep learning (DL) is a subtype of ML which uses multilayered artificial neural networks (ANNs) to derive high-level features from input data (similar to neuronal networks) and to perform complex tasks in medical imaging. Specifically, DL is useful in classification tasks and in automatic feature extraction, where it is able to solve the issues of partial detectability and feature accessibility when trying to extract information from these valuable data sources. The use of multilayered convolutional neural networks (CNNs) improves DL robustness by mimicking human neuronal networks in the training phase. If applied to unlabelled data, the automatic process of learning relies on image features being automatically clustered based on their natural variability. Due to the challenge of achieving completely unsupervised instruction, complex learning models are most commonly implemented with a degree of human supervision. The performance of CAD can be increased using ML and CNNs. CAD systems utilizing ML can be trained on a dataset from a representative population and then identify the features of a single lesion in order to classify it as normal or abnormal [16]. Algorithms based on statistics are the major focal point of both supervised and unsupervised learning [17]; however, there are important variations. Classification (i.e., categorizing an image as normal or abnormal based on labels provided in the training data) and regression (observing or finding new categories using inference on training sets) are the two main applications of supervised learning. Unlike supervised learning, unsupervised models use unlabelled/unclassified data. As a result, latent pattern recognition is accomplished through dimensionality reduction and feature clustering [17]. To determine the usefulness of this classification process, it must first be validated. The capacity to link basic diagnostic patterns and features of medical image modalities to a specific pathological and histological subtyping has led to the area of radiomics by merging DL-based image processing with clinical, and when suitable, pathological/histological data [18,19,20].

### 2.2. Radiomics

Radiomics is a recent translational field in which a variety of properties, including geometry, signal strength, and texture, are extracted from radiological images in order to record imaging patterns and categorize tumor subtypes or grades. Radiomics is typically utilized in systems with various variants for prognosis, monitoring, and determining how well a treatment is working [19,21]. “Images are more than pictures; they’re data.” The basic concept of radiomics is beautifully illustrated by Robert Gillies and colleagues’ intuitive and precise description [4]. Radiomics is classified into two types, namely, feature-based and DL-based radiomics, and is commonly used to analyze medical images at a low computational cost. While clinical evaluations are subject to inter-observer variability, results using radiomics are more accurate, stable, and reproducible, as automated radiomic characteristics are either generated statistically from ML-based complicated computational models during the training phases (DL-based) or computed using mathematical methods (feature-based). However, in order to achieve a correct diagnosis, the input data must be of high quality, with accurate labels (in the case of supervised learning) or a population-representative sample (in the case of unsupervised learning) [22].

#### 2.2.1. Feature-Based Radiomics

In order to perform feature-based radiomics, a segmented volume of interest (VOI) for 3D data or region of interest (ROI) for 2D data is used. In order to avoid overfitting (an erroneous reliance on clinically irrelevant features), feature selection algorithms examine a subset of certain features after feature extraction and create robust prediction models. Overfitting usually occurs in datasets that are too homogeneous and lack enough representations of the target diseases. As a result, the chosen features may not be valid when applied to different data sets. Selecting features from heterogeneous and representative datasets decreases the likelihood of selecting narrowly applicable features and reduces the risk of overfitting. Feature-based radiomics does not necessitate large datasets, as the features are typically defined after only a short computation time. Notably, the majority of extracted features are very complex, and often do not correlate to recognizable pysiological or pathological features, limiting model interpretability. The feature-based radiomics process contains the key processing stages listed below.

Image pre-processing is a basic step in radiomics in which useful features are extracted. The primary goal of radiomics is the generation of quantitative features from radiology images [4,22,23,24,25,26]. The generated features and models should be both repeatable and general, particularly when using multi-variate multi-model data (i.e., with different scanners, modalities, and acquisition protocols), which is the case for most medical imaging centers. Several pre-processing steps are necessary to achieve these objectives. Correcting inhomogeneities in MRI images, reducing noise, spatial resampling, spatial smoothing, and intensity normalization are all common pre-processing steps for radiomic analyses [19,27,28].

Tumor segmentation is another extraction step in feature-based radiomics. Segmentation of MR or CT images for different types of tumors is typically performed manually, either in preparation for ML model development or in clinical practice for planning of radiotherapy or volumetric evaluation of treatment response [26]. It takes a great deal of effort to carry out 3D manual segmenting of lesions that have necrosis, contrast enhancement, and surrounding tissue. The contours have a direct effect on the radiomics analysis results, as the segmented ROIs define the input for the feature-based radiomics process. To handle this challenge, ML approaches are being developed for automatic tumor localization and segmentation [29,30].

Feature extraction from medical images may lead to several quantitative traits, the majority of which are tumor heterogeneity. Although many features can be extracted, features are typically divided into four subgroups:Shape characteristics, such as sphericity, compactness, surface area, and maximum dimensions, reflect the geometric characteristics of the segmented VOI [26].First-order statistical features or histogram-based features describe how the intensity signals of pixels/voxels are distributed over the segmented ROI/VOI. These features neglect the spatial orientation and spatial relationship between pixels/voxels [23].Second-order statistical features or textural features are statistical relationships between the signal intensity of adjacent pixels/voxels or groups of pixels or voxels. These features serve to quantify intratumoral heterogeneity. Textural features are created by numerically characterizing matrices that encode the exact spatial connections between the pixels/voxels in the source image. The gray-level co-occurrence matrix (GLCM) [31] is the most widely used texture analysis matrix. The GLCM shows how many times two intensity levels appear in adjacent pixels or voxels within a given distance and in a defined direction. Multiple textural characteristics, including energy, contrast, correlation, variance, homogeneity, cluster prominence, dissimilarity, cluster inclination, and maximum likelihood, can be measured using the GLCM. The difference in intensity levels between one pixel/voxel and its 26-pixel 3D neighborhood is represented by the neighborhood gray-level different matrix (NGLDM). For each image intensity, the gray-level run length matrix (GLRLM) encodes the size of homogeneous runs [32]. Long-run emphasis (LRE), short-run emphasis (SRE), low gray-level run emphasis (LGRE), run percentage (RP), and high gray-level run emphasis (HGRE) can all be derived from the GLRLM. There are other matrices that capture pixel-wise spatial relationships and can be used to compute additional texture-based features [31].Higher-order statistical features are quantified using statistical methods after applying complex mathematical transformations (filters), such as for pattern recognition, noise reduction, local binary patterns (LBP), histogram-oriented gradients, or edge enhancement. Minkowski functionals, fractal analysis, wavelet or Fourier transforms, and Laplacian transforms of Gaussian-filtered images (Laplacian-of-Gaussian) are examples of these mathematical transformations or filters [28].

Feature selection is a helpful step used to refine the set of extracted features. For implementation of image-based models for prediction and prognosis, the extracted quantitative features might not have equal significance. Redundancy, overly strong correlation, and feature ambiguity can cause data overfitting and increase image noise sensitivity in the dependent predictive models. Overfitting is a methodological error in which the developed model is overly reliant on features specific to the radiological data used in the training process (i.e., noise or image artifacts) rather than features of the disease in question. Overfitting results in a model with deceptively high classification scores on the training dataset and weak performance on previously unseen data. One way to reduce the risk of overfitting is to employ feature selection prior to the learning phase [9]. Supervised and unsupervised feature selection techniques are widely used in radiomics. Unsupervised feature selection algorithms disregard class labels in favor of eliminating redundant spatial features. Principal component analysis (PCA) and cluster analysis are widely used techniques for this type of feature selection [19]. Despite the fact that these approaches minimize the risk of overfitting, they seldom yield the best feature subset. Supervised feature selection strategies, on the other hand, consider the relations between features and class labels, which results in the selection of features based on how much they contribute to the classification task. In particular, supervised feature selection procedures select features that increase the discrimination degree between classes [26]. For supervised feature set reduction, there are three common methods:Filter methods (univariate methods) examine how features and labels are related without taking into account their redundancy, or correlation. Minimum redundancy maximum significance, Student’s *t*-test, Chi-squared score, Fisher score, and the Wilcoxon rank sum test are among the most commonly used filter methods. While these feature selection methods are widely used, they do not take the associations and interactions between features into account [33,34].Wrapper methods (multivariate methods), known as greedy algorithms, avoid the filter method constraint by looking at the entire space of features and considering the relationships between each feature and other features in the dataset. A predictive model is used to evaluate the output of a group of features. The consistency of a given technique’s output is used to test each new subset of features. Wrapper approaches are computationally intensive, as they strive to find the best-performing functional group of features. Forward feature selection, backward feature exclusion, exhaustive feature selection, and bidirectional search are all examples of wrapper methods [33,34].Embedded approaches carry out the feature selection process as part of the ML model’s development; in other words, the best group of features is chosen in the model’s training phase. In this way, embedded approaches incorporate the benefits of both the filter and wrapper methods. Embedded approaches provide more reliability than filter methods, have a lower execution time than wrapper methods, and are not very susceptibility to data overfitting, as they take into account the interactions between features. The least absolute shrinkage and selection operator (LASSO), tree-based algorithms such as the random forest classifier, and ridge regression are examples of commonly used embedded methods [33,34].

Model generation and evaluation is the final step in feature-based radiomics. Following feature selection, a predictive model can be trained to predict a predetermined ground truth, such as tumor recurrence versus tissues changes related to treatment. The most commonly used algorithms in radiomics include the Cox proportional hazards model for censored survival data, support vector machines (SVM), neural networks, decision trees (such as random forests), logistic regression, and linear regression. To avoid overfitting of ML models when using supervised methods, datasets are usually split into training and validation subsets to ensure that these subsets maintain a sample distribution similar to the class distribution; this is particularly important for small or unbalanced datasets. After training and validating the model, a previously unseen testing subset of data is introduced to test the model. Optimally, the testing data should be similar to the actual data that the model will work on in real clinical settings, and should be derived from a different source (e.g., a different institution or instrument) than the training data. As a consequence, when testing a model’s performance, robustness, and reliability, the testing dataset is the gold standard. Alternatively, statistical approaches such as cross-validation and bootstrapping can be used for model output estimation without using an external testing dataset, particularly for small datasets.

#### 2.2.2. DL-Based Radiomics

ANNs that mimic the role of human vision are used in DL-based radiomics to automatically generate higher-dimensional features from input radiological images at various abstraction and scaling levels. DL-based radiomics is particularly useful for pattern recognition and classification of high-dimensional and nonlinear data. The procedure is radically different from the one described above. In DL-based radiomics, medical images are usually analyzed using different network architectures or stacks of linear and nonlinear functions, such as auto-encoders and CNNs, to obtain the most significant characteristics. With no prior description or collection of features available, neural networks automatically identify those features of medical images which are important for classification [35]. Across the layers of a CNN, low-level features are combined to create higher-level abstract features. Finally, the derived features are used for analysis or classification tasks. Alternatively, the features derived from a CNN can be used to generate of other models, such as SVM, regression models, or decision trees, as is the case when using feature-based radiomic approaches. Feature selection is seldom used, because the networks produce and learn the critical features from the input data; instead, techniques such as regularization and dropout of learned link weights are used to avoid overfitting. Compared to feature-based radiomics, larger datasets are required in DL-based radiomics due to the high correlation between inputs and extracted features, which limits its applicability in many research fields that suffer from restrictions in data availability (such as neuro-oncological studies). Notably, the transfer learning approach can be used to circumvent this obstacle by employing pre-trained neural networks for a separate but closely related purpose [36]. Leveraging the prior learning of the network can achieve reliable performance even with limited data availability. These two types of radiomics and their different steps are illustrated in Figure 2.

### 2.3. Evaluation Metrics in AI and Radiomics Techniques

To evaluate the performance of AI and radiomic techniques, a number of different metrics can be used. With respect to model evaluation, TP denotes true positive, TN denotes true negative, FN denotes false negative, and FP denotes false positive. The following metrics are used over a number of repetitions to evaluate AI and radiomics techniques:Total Score: evaluation of different segmentation measures, namely, the overlap error, the relative absolute volume difference, and the surface distance (in terms of mean, RMS, and maximum).Dice similarity coefficient (Dice score) = 2TP/(2TP+FP+FN)Accuracy = (TP+TN)/(TP+TN+FP+FN)Sensitivity = TN/(TN+FP)Specificity = TP/(TP+FN)Area under the curve (AUC): The area under the receiver operating characteristics (ROC) curve, which connects the true positive rate (sensitivity) and the false positive rate (1-specificity); the AUC value ranges from 0 to 1, with 1 being the best performance.

### 2.4. Clinical Application of AI and Radiomic Techniques in Liver Cancer

Hepatocellular carcinoma (HCC), the most prevalent form of liver cancer, develops in the the hepatocytes, the primary type of liver cell. Hepatoblastoma and intrahepatic cholangiocarcinoma are two significantly less frequent kinds of liver cancer. Because HCC is the most prevalent form of liver cancer, the next sections focuses on the different AI and radiomics techniques for HCC segmentation, detection, and management.

## 3. Segmentation of Hepatic Focal Lesions

AI-based automatic segmentation of hepatic focal lesions is preferred to manual or semi-manual segmentation, as the latter methods are liable to personal variations and are far more time-consuming. Unfortunately, different models proposed for liver and HCC automatic segmentation are less accurate than human-based methods. This is attributed to the heterogeneity of the background hepatic parenchyma in cases of HCC in a cirrhotic liver, which is the case for most HCC patients. The variable shape, size, density, and location of hepatic lesions in CT or MR images contribute to inaccuracies in automated segmentation methods. On CT, Hepatic tumors can appear hypodense, hyperdense, or be of mixed density, with foci of hemorrhage, calcifications, and necrosis. In addition, there are different patterns of enhancement after contrast injection in triphasic studies [37]. In addition to variability related to tumor type, grade, stage, and imaging parameters (machine, imaging protocol, type and timing of contrast), the appearance of focal hepatic lesions is subject to individual variations from patient to patient. Older algorithms for automatic segmentation have used adaptive thresholding, region growing, level set techniques [38,39,40,41], Grassmannian manifolds [42], and shape parameterization [43]. More recent algorithms use DL methods to achieve more accurate results. Christ et al. [44] applied two cascaded U-net models to perform liver and lesion segmentation in two separate processes within a liver bounding box. Then, the final outcome was refined using a 3D conditional random field (CRF). In 2017, the liver tumor segmentation (LiTS) challenge was organized. The top winning algorithms were based on focal neural networks (FNN), which are designed to learn features from data in an automatic way. A pair of U-net-like models with long and short skip connections were proposed by the first-round winner. The initial model was only applied to coarse liver segmentation, while the second network was trained to segment both liver and tumors in one step. The two models worked in 2.5D by receiving five consequent slices to segment the middle one, and the final output was presented as 3D context information [45]. Other promising methods were based on training two networks for liver and tumor segmentation and on developing 3D information by training a 3D H-DenseUNet utilizing original image data in addition to features from a 2D network [46,47]. More recently, Chlebus et al. [48] designed an automatic segmentation method based on a 2D CNN with an object-based postprocessing step. Their algorithm utilized two models, a voxel-level one and an object-level one, that in turn reduced false positive results by about 85% in comparison to other raw neural network outputs. Another more complex algorithm based on a 3D dual-path CNN was introduced by Meng et al. [49]. They used CRF to remove the false segmentation points in the segmentation results in order to refine the final segmentation and improve the accuracy. Table 1 summarizes the state-of-the-art hepatic lesions segmentation systems.

Table 1 illustrates that the automatic segmentation of hepatic lesions is performed using two main categories of techniques, namely, feature-based and DL-based. Extracting numerous amounts of low- and high-level features through multiple covolutional layers using DL-based techniques [44] leads to a significant improvement in segmentation performance; however, it is limited by being suitable only for larger data cohorts. Recently, many research studies have worked to address this issue using several techniques, including transfer learning and semi-supervised deep learning.

## 4. Detection and Differentiation of HCC from Other Hepatic Masses

The application of AI in the field of abdominal imaging provides a potent tool to help less experienced radiologists diagnose HCC more accurately and discriminate it from HCC mimics using specific CT or MR radiomics. Textural features are further classified into statistical features (which deal with the distribution of grayscale values), model-based (concerned with irregularity of the area), and transform-based (which turn spatial data into frequency) [50]. In clinical practice, statistical features are the most commonly used [26,51,52,53]. Nie et al. [54,55] designed a nomogram based on CT radiomics to differentiate HCC from focal nodular hyperplasia (FNH) and hepatocellular adenoma (HA) in the normal non-cirrhotic liver with high accuracy. Yasaka et al. [35] used a CNN to differentiate HCC and other malignant lesions with radiological features not typical for HCC from other indeterminate lesions, hemangiomas, and cysts, achieving a median accuracy of 0.84. Wu et al. [28] extracted MR radiomics from pre-contrast images (in and out of phase T1WI, T2 WI, and DWI) and then used four different types of classifiers to discriminate hepatic hemangioma (HH) from HCC. Their results showed that their model had a diagnostic performance that outperformed a less experienced radiologist (with two years of experience as an abdominal radiologist), and nearly equaled to the performance of an experienced one (ten years of experience). In the cirrhotic liver, differentiation between HCC and cirrhotic nodules was achieved with high accuracy in the work of Morkos et al. [56]. Ponnoprat et al. designed a two-step model to differentiate HCC from intrahepatic cholangiocarcinoma (CC) with 88% accuracy [57]. Another more recent study [58] found that MRI radiomics performed better than CT radiomics in the discrimination of combined HCC-CC from either HCC or CC. They reported that post-contrast MRI, non-contrast CT, and the portal venous phase of triphasic CT had the best results in differentiating HCC from CC or combined HCC-CC. One interesting study [59] used a convolutional dense network (CDN) to evaluate the accuracy of three models of CT imaging in differentiating HCC from other focal hepatic lesions. They found that triphasic CT without pre-contrast images had the highest accuracy (85%). Another study found that using multi-phase CDN models yielded high accuracy and specificity in differentiating HCC, metastasis, non-inflammatory benign focal masses, and hepatic abscesses from each other [60]. Regarding MRI-based approaches, two consecutive studies [61,62] developed CNN models and highlighted feature maps based on MRI radiomics to differentiate between six categories of hepatic masses. They achieved higher sensitivity and specificity than experienced radiologists, especially in the discrimination of HCC from other hepatic observations. Other researchers [63] designed a DL system that collects data from clinical sources (clinical documentation and laboratory results), non-contrast MRI, and post-contrast MRI images. Their results showed that the diagnostic accuracy of the DL system was equivalent to the performance of three experienced radiologists who were asked to classify malignant liver masses into seven categories. They suggested that because their design performed well with non-contrast images, future imaging protocols could avoid the use of intravenous contrast. Another study proposed a deeply supervised cross-learning model, which was able to significantly improve the characterization of HCC based on non-contrast MRI images [64]. A novel article concluded that feature-based radiomics along with multiphasic post-contrast MRI could detect microscopic juxta-early HCC [65]. A summary of recent studies utilizing AI in the diagnosis of HCC is shown in Table 2.

As shown in Table 2, depending on the aim and research question, each research study develops its own methodology, selects unique characteristics, or makes use of various sets of CNN and classifiers; thus, there is a lack of scalability. Radiomics-based diagnostic models are developed using a variety of techniques, with no specific protocol followed. Hence, developing an application that merges DL and clinical biomarkers in a way that achieves generalization and adaptability is of great importance in the field of differentiation of hepatic lesions. Development of such an application would require a very large dataset in order to apply sufficient training/testing.

## 5. Managment of HCC

In this section, different clinical applications of AI and radiomics techniques for the management of HCC are outlined, including the prediction of HCC histopathology, monitoring of locoregional therapeutic response, prediction of response to hepatic resection and systemic treatment and/or immunotherapy, and the prediction of survival of HCC patients.

### 5.1. Prediction of HCC Histopathology

#### 5.1.1. Microvascular Invasion

Microvascular invasion (MVI) of HCC is considered an important predictor of tumor recurrence; it is defined as the presence of tumor cells in the main portal vein or one of its branches, hepatic veins, or large capsular veins [66]. Several studies have attempted to find a correlation between CT radiomics and MVI [67,68,69,70,71]. These studies have found that a combination of radiological scores (LiRADS) and clinical data (such as prior infection with hepatitis B virus) with data obtained by nomogram yields the highest accuracy (88%). A number of researchers have designed models combining data from peri-tumoral and tumoral areas to predict MVI, as vascular invasion usually occurs at the tumor margins [67,68,70]. Jiang et al. [72] designed models based on eXtreme Gradient Boosting (XG Boost) and DL. Their first model combined radiomic features, imaging data, and clinical data (RRC), while the second was a three-dimensional CNN model (3D CNN). With an accuracy of 85.2%, the 3D CNN model outperformed the feature-based model (accuracy = 84.0%). Another study [73] compared different radiomics approaches, and reported that LASSO+GBDT models had the best diagnostic performance for prediction of MVI. A recent meta-analysis concluded that while both ML and non-ML methods are beneficial in predicting MVI, ML models have shown better overall results [74]. One recent study proposed a deep learning framework composed of a 3D CNN and a loss function deeply supervised net that extracted deep features from multi-phasic contrast-enhanced MRI. Their combined model showed good performance (AUC = 0.926) in predicting MVI when using data derived from all phases [75]. Another study [76] reported encouraging results from analysis of the histogram of the peritumoral region in post-contrast arterial phase MRI images to predict MVI in patients with single HCC lesions. On the contrary, Dai et al. [77] concluded that a CNN model created using radiomic features derived from the hepatobiliary phase of enhanced MRI images using a gradient boosting decision tree (GBDT) classifier had the highest predictive accuracy regarding MVI. They found that GBDT outperformed other classifiers (SVM, logistic regression with the least absolute shrinkage and selection operator (LASSO), minimum redundancy maximum relevance (mRMR)). Wang et al. [78] found that deep features extracted from diffusion-weighted images (DWI) with a b-value of 600 were more accurate than those extracted at b0, b100, or individually from the apparent diffusion coefficient (ADC) map. The highest prediction performance was reached when combining data from DWI of the three b-values and ADC map together. On the other hand, Meng et al. [79] reported that there was no significant difference in the accuracy of radiomics extracted from CT or MRI in pre-operative prediction of MVI in patients with single HCC, apart from those with masses 2–5 cm, for whom MRI showed higher accuracy. Table 3 presents a summary of the aforementioned studies.

#### 5.1.2. HCC Grade and Molecular Signature

The pre-operative prediction of tumor grade is valuable in the prediction of long-term survival as well as therapeutic response. It may even play role in treatment planning, as high-grade tumors may require more aggressive treatment and a wider safety resection margin [80,81,82,83]. Mao et al. [84] designed an ML model to predict high-grade HCC based on first-order, second-order, higher-order, and shape features derived from arterial and venous phases of triphasic CT utilizing recursive feature elimination and eXtreme Gradient Boosting (XGBoost). Their results showed comparable accuracy between post-contrast CT-derived radiomics and clinical factors, with the best prediction achieved when using combined models (AUC = 0.8014). On the other hand, Wu et al. found that MR radiomics derived from non-contrast images outperformed other clinical biomarkers in differentiating high-grade from low-grade HCC (AUC = 0.74 vs. 0.60, respectively), while a combination of MR radiomics and clinical/laboratory biomarkers showed the best performance (AUC = 0.8) [85]. Another study found that using an artificial neural network (ANN) was more accurate than LR in classifying low- and high-grade HCC based on fradiomic features extracted from the arterial phase, hepatobiliary phase, or both [86]. A group of researchers created a 2D CNN model based on features extracted from 2D-log maps generated from DWI images of different b-values. They found that their model performed better in grading HCC than other models that derived features from ADC maps directly or from individual DWI of the 0, 100, and 600 b-values separately [87]. The pre-operative detection of dual-phenotype HCC (DPHCC) is crucial for treatment plans, as there is increasing scientific evidence that the expression of CK19 is associated with more aggressive tumor behavior and resulting higher rates of invasion, proliferation, migration, and tumor recurrence [88,89]. One study used radiomics derived from gadolinium-ethoxybenzyl-diethylenetriamine pentaacetic acid (Gd-EOB-DTPA)-enhanced images to detect DPHCC pre-operatively. They applied four types of classifiers (logistic regression (LR), SVM, K-nearest neighbor (KNN), and multi-layer perception (MLP)), and found that a combination of features extracted from multiple phases along with applications of multiple classifiers had the strongest diagnostic power [90]. Another study reported significant correlation between tumor grade and expression of CK-7, CK19, and GPC-3 based on radiomic features extracted from susceptibility-weighted images (SWI). No such correlation was found for MVI [91]. Wang et al. [92] found that DL models based on feature radiomics derived from arterial and hepatobiliary phases of MRI could be considered as reliable biomarkers of CK-19 expression. A multi-center study [93] proposed a radiomic model with high accuracy in pre-operative prediction of CK-19 expression. Their model was based on features extracted from multi-sequence gadoxetic acid-enhanced MR.

One of the most favorable biomarkers used in the prognosis of HCC is k-67 expression. A sophisticated study used Gd-EOB-DTPA radiomics in combination with clinical risk factors to pre-operatively predict the expression of k67 in HCC. They picked features from five imaging sequences (arterial phase, portal venous phase, delayed phase, hepato-biliary phase, and T2-WI) and followed this with the application of LASSO to build a radiomic score. Their results showed that the combination of radiomic scores obtained from the arterial phase and serum AFP yielded the best predictive performance [94]. The different models discussed above are summarized in Table 4.

### 5.2. Monitoring of Locoregional Therapeutic Response

Locoregional therapy (LRT) of HCC includes several maneuvers, such as percutaneous ethanol injection (PEI), micro-wave ablation (MWA), radiofrequency ablation (RFA), cryo-ablation, trans-arterial embolization (TAE), drug-eluting bead trans-arterial chemo-embolization (DEB-TACE), trans-arterial radio-embolization (TARE), stereotactic body radiotherapy (SBRT), and trans-arterial chemoembolization (TACE). These maneuvers aim to improve prognosis and delay the progress of the neoplasm. They can act as a bridge before a permanent cure through either surgical excision or liver transplant [95,96]. It is crucial to determine whether these procedures successfully eradicate the tumor or whether there is active tumor tissue remaining, as the latter necessitates further treatment sessions. This is a challenging task, as each procedure has its own peculiar post-therapeutic changes, including necrosis, hemorrhage, or retained embolic materials. AI has been introduced to assist radiologists in the detection of volumetric measurements of viable tumor tissue as well as prediction of the response. For usual radiological assessment, complete response after RFA, MWA, and PEA necessitates absent enhancement. However, thin peripheral enhancement can persist for months after cryoablation, TAE, TACE, DEB-TACE, TARE, and SBRT. Nodular enhancement can be present for a few months after TARE, while tumors treated with SBRT may show what is called a pseudo-progression in both size and enhancement for about three months post-therapy. This must be taken into consideration, and should not reported as a persistent viable tumor [97,98,99,100]. It is obvious that accurate assessment of tumor response to LRT necessitates great experience, as there are many potentially ambiguous imaging findings. In addition, there is a large segment of patients who undergo more than one type of LCT. Considering these complexities, the application of AI to this problem could be valuable.

#### 5.2.1. Prediction of Response to Ablation Therapy (MWA & RFA)

An et al. [101] designed a deep learning-based deformable image registration (DIR) technique to assess the ablation margin in HCC treated by MWA and correlate it with local tumor progression (LTP). They compared post-contrast MRI images dated one month prior and three months after MWA to measure the ablative margin, which was defined as the distance between the original tumor and the deformed ablated region. They used an unsupervised landmark-constrained CNN to correct misalignments between images. They declared that an ablative margin equal to or less than 5 mm was an independent risk factor for LTP. Hu et al. [102] developed a nomogram consisting of five features (AFP, tumor number, peri-tumoral enhancement in arterial phase, hypointense SI in hepatobiliary phase of Gadoxetic acid-enhanced MRI) to predict early recurrence in post-ablative therapy for HCC. Their nomogram had high accuracy (AUC = 0.843). Linag et al. [103] developed an SVM that was able to predict the recurrence of HCC in patients who were treated with RFA with high specificity and accuracy (86% and 82% respectively). This approach could be used to select patients with a high risk of recurrence for placement in a close follow-up protocol. Another recent study involved developing a DL model based on radiomic features extracted from contrast-enhanced ultrasound (US) images of patients who had RFA or local hepatic segment resection as a treatment for HCC. They tested their model to predict progression-free survival (PFS) and optimize treatment selection in patients having very early or early-stage HCC. Their results showed that feature radiomics combined with clinical biomarkers could predict PFS and help in treatment selection to improve patient survival [104]. Table 5 summarizes these different studies targeting the prediction of response to ablation therapy.

#### 5.2.2. Prediction of Response to TACE

Several studies have reported the efficacy of CT and MRI radiomics in predicting tumor response after TACE [105,106,107,108,109,110,111,112,113,114,115,116]. Morshid et al. [106] used quantitative features from pre-operative CT in combination with clinical data to build an ML predictive model that showed high accuracy (74.2%) in predicting tumor recurrence after TACE. Similarly, Meng et al. [107] found that a combined clinical/radiomics model had the best predictive performance. Peng et al. [108] created a DL model to predict complete/partial response or stable/progressive disease in 562 patients who received TACE as a treatment for intermediate stage HCC. They extracted data from post-contrast CT, then applied transfer learning techniques through a residual CNN (ResNet50). Their ML training model showed high accuracy (84.3%), while the other two validated cohorts had accuracies of 85.1% and 81.8%. Sun et al. [111] designed a predictive model based on MRI radiomics and found that the highest predictive accuracy was achieved when radiomic features were extracted from multiparametric MRI (AUC = 0.8), while models derived from radiomics of individual sequences had the following results in descending order of accuracy: DWI b0 (AUC = 0.786), DWI b500 (AUC = 0.729), T2-WI (AUC = 0.729), and finally ADC (AUC = 0.714). Several studies [112,113,114,115,116] have found that predictive models based on MRI radiomics are comparable to or even outperform those based on clinical biomarkers, with combinations of clinical data and MR radiomics having the best predictive performance. Two recent studies have reported promising results in monitoring response to combined MWA and TACE [117], high-intensity focused US, and TACE [118] based on analysis of MR textural features.

### 5.3. Prediction of Response to Hepatic Resection

Regardless of the continuous progress of locoregional therapeutic modalities, resection remains the gold standard ideal treatment for HCC [119]. However, as HCC is an aggressive tumor, even post-resection patients have a high recurrence rate approaching 70% [120]. Thus, a predictive tool with high accuracy is necessary to evaluate patients pre-operatively and select those with a low risk of local recurrence who can benefit from relatively invasive and expensive procedures such as local hepatic resection. Additionally, such tools can help in the planning of proper surveillance programs for high-risk groups [121,122]. Researchers have proposed predictive models with high accuracy based on combinations of CT radiomics and clinical risk factors to assess short- and long-term survival after surgical resection of HCC [123,124]. One such study found that random survival forest (RSF) was able to extract eight histogram textural features related to disease-free survival and ten features related to overall survival (OS) [125]. Interestingly, several studies have reported correlation between different textural features derived from pre- and post-contrast CT of resectable HCC and survival [126,127,128]. Shan et al. [129] used peritumoral CT radiomics (2 cm around the original tumor) to predict the recurrence of HCC after hepatectomy or ablation. Their study showed that peritumoral CT radiomics were more accurate than tumoral CT radiomics in the prediction of tumor recurrence. On the other hand, Kim et al. [130] used peritumoral MR radiomics (3 mm from the tumoral edge) in building a pre-operative predictive model that showed similar results to post-operative clinic-pathologic model. A multi-center study proposed a pre-operative and a combined pre- and post-operative prognostic model using multivariable Cox regression analysis. Their pre-operative model was composed of radiomic signatures derived from post-contrast CT, tumor number, and serum AFP level, while the post-operative model added satellite nodules and MVI to the aforementioned factors. Models created with radiomics had better prognostic performance than those depending only on clinical predictors [131]. Similarly, another multicenter study found that a combined model composed of clinical data (AFP, albumin-bilirubin grade, tumor margin, and liver cirrhosis) and three-feature radiomic signatures achieved high prognostic performance in predicting recurrence in post-resection patients [132]. Wen et al. developed a combined model with high accuracy (AUC = 0.981) composed of clinical risk factors, radiologic features, and radiomic scores to predict early and late tumor recurrence in small HCC (3 cm and less) treated with either local resection or RFA [133]. Similar results have been reported by other researchers, albeit with lower accuracy [134,135,136,137,138,139].

### 5.4. Prediction of Response to Systemic Treatment and/or Immunotherapy

Unresectable HCC is currently treated by targeted therapeutic agents as Sorafenib (oral multikinase inhibitor of the vascular endothelial growth factor receptor, the platelet-derived growth factor receptor) and Lenvatinib (inhibitor of VEGF receptors 1–3, FGF receptors 1–4, PDGF receptor α, RET, and KIT) in addition to immunotherapeutic agents, resulting in impressive clinical response [140,141,142,143,144,145,146,147,148,149]. The response of tumors to immunotherapy is highly dependent on the immune status of the tumor itself and its immunoprofiling [150]. Liao et al. [151] developed a radiomics-based biomarker derived from post-contrast-enhanced CT that proved to be an accurate predictor of tumor cell infiltration with CD8+T. Similarly, Yuan et al. [152] developed a CT radiomic/clinical nomogram with high performance (AUC = 0.883) to predict anti-PD-1 treatment efficacy in patients with advanced HCC. They incorporated eight CT radiomic features derived from enhanced images of the whole tumor and peritumoral region, in addition to two clinical factors (tumor embolus and ALBI grade). Hectors et al. [153] found that MRI feature radiomics can predict HCC immunoprofiling well, and might help in risk stratification. Recent reports have found that higher enhancement of intrahepatic HCC nodules in the hepatobiliary phase of Gd-EOB-DTPA-enhanced MRI may reflect the activation of the Wnt/β-catenin signaling pathway. Furthermore, it has been reported that the enhancement values could be predictors of response to immune-checkpoint inhibitors (ICI) treatments [154,155,156].

### 5.5. Prediction of the Survival of HCC Patients

Various factors appear to influence the survival of HCC patients, with the most recognizable factors including the initial status of the liver, presence of co-morbidities, characteristic radiological and pathological features, and the presence or absence of certain biological markers [157]. The traditional way of predicting survival is through survival analysis and Cox proportional hazard models [158,159,160]. Recently ANNs have been used to assess different clinical-pathological outcomes in patients with hepatectomy or other locoregional procedures as treatment for HCC. An ANN is composed of an input layer that receives data, multiple hidden layers to analyze the data, and an output layer that provides the result. An ANN is capable of showing input and output data in pairs, which improves reliability [161]. Hamamoto et al. [162] used a neural network to predict the outcome in eleven patients who had undergone hepatectomy as treatment for HCC, and their method showed 100% successful prediction. Other researchers have found ANNs to have better performance in predicting the survival-free interval [163] and in-hospital mortality [164,165,166] post-hepatectomy as compared to traditional logistic regression models. In line with the results described in previous sections of this article, several studies have shown that models derived from CT or MRI radiomics perform better than clinical nomograms in prediction of tumor recurrence, and could therefore be a useful tool for individual patient prognosis and the selection of patients who might benefit from LRT, hepatectomy, or systemic therapy [124,134,137,167,168,169,170,171].

Throughout this section, it is notable that the aforementioned studies concerned with the management of HCC show that applying DL-based radiomics leads to higher performance. On the other side, these approaches are limited by the fact that they need to be connected to specific clinical markers in order to afford better understandability of disease. Moreover, they are costly in terms of time and computing resources.

## 6. Limitations and Future Perspectives

One of the most important limitations within the literature is that most of the aforementioned studies were retrospective and included a relatively small number of patients. Another limitation is the lack of generalization, as each study creates its own methods, picks different features, and utilizes different sets of CNN layers and classifiers. There is no fixed protocol to follow, and multiple different software applications are used to extract radiomics features. Indeed, a large public open-access database, more multicenter studies, and clinical trials would promote more validated techniques and procedures. One recent multicenter study [172] standardized a set of 169 radiomics features, which in turn facilitated the validation of radiomics software. Another limitation that may hinder the wider application of AI in routine imaging is the high cost of its complex processes. To build a proper diagnostic or predicting model, images must pass through segmentation, feature extraction, and application of classifiers, then modeling, which requires specialized software packages and specialized personnel able to manage these software applications. Another important issue to be highlighted is that most radiologists are not currently familiar with the complexities of AI-based software.

However, AI-based models are becoming more widely used in the management of HCC and many other illnesses to decrease interobserver variability and bias. Necessarily, this requires wide-scale cooperation and effort to create the enormous high-quality datasets required to train such models. Such cooperation, efforts, and larger datasets can enhance the general application of the powers of AI for utilization with clinical biomarkers, and could be used for different applications in the field of HCC.

## 7. Conclusions

AI, ML, and DL have been used to develop models that can improve the diagnosis and management of HCC. In this article, we cover several techniques: detection and differentiation of HCC from other hepatic masses, prediction of histological grading and MVI, monitoring of therapeutic response, and prediction of survival. From approximately 130 studies, our conclusions show initial results for hepatic lesion segmentation with dice scores ranging from 67% to 94%. For detection and differentiation of HCC from other hepatic observations, the included AI-based models yield accuracies in the range of 77% to 91.9%. Moreover, microvascular invasion can be predicted with an AUC ranging from 0.76 to 0.90 using different approaches. Additionally, the included AI-based studies achieve high performance in the prediction of HCC grade, therapeutic response, and survival. Yet, these complex techniques lack the validation and standardization necessary for their application in routine everyday work.

The performance of AI in HCC-related tasks cannot yet be regarded as a gold standard, despite its potential. However, AI models are becoming more reliable and broadly applicable and attaining the capacity to anticipate novel scenarios, demonstrating that AI can be used to design more precise and personalized management models that diminish subjectivity-related biases.

## Figures and Tables

**Figure 1 cancers-14-06123-f001:**
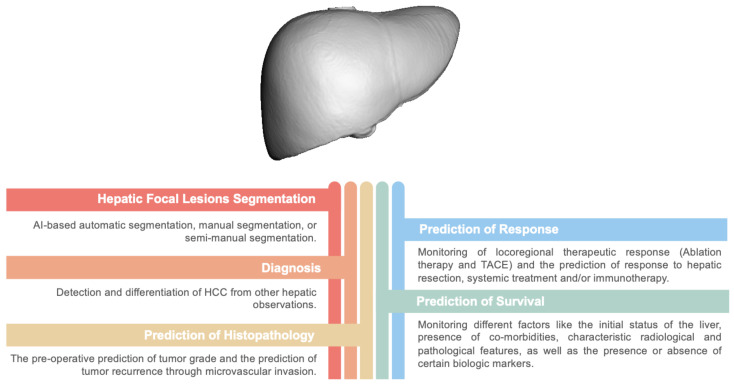
A summary of different applications of Artificial Intelligence and Radiomics in the field of HCC.

**Figure 2 cancers-14-06123-f002:**
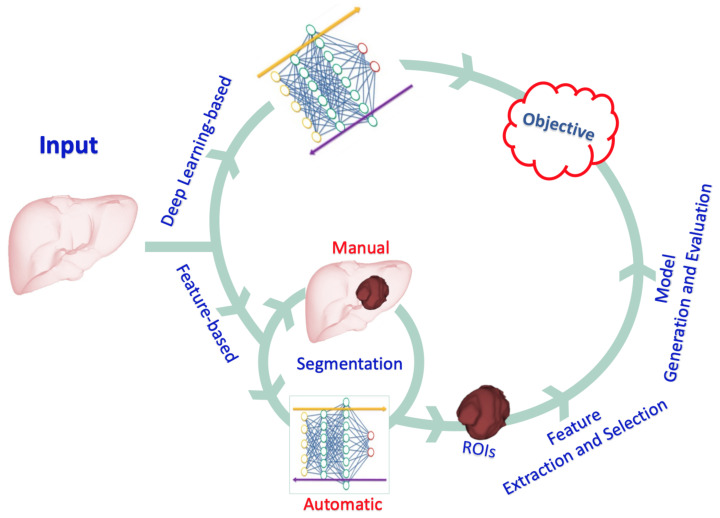
An illustration of the different types of radiomics and the steps involved.

**Table 1 cancers-14-06123-t001:** Segmentation of Hepatic Focal Lesions.

Study	Method	CT Data	Performance
Hame [38]	Simple thresholding Fuzzy clustering and deformable model	10 subjects	Qualitative evaluation
Smeets et al. [39]	Supervised statistical pixel classification	10 subjects	Total score = 69%
Choudhary et al. [40]	Adaptive multi-thresholding	10 subjects	Total score = 70%
Moltz et al. [41]	Adaptive thresholding Shape analysis	10 subjects	Total score = 72%
Kadoury et al. [42]	Appearance analysis using discriminant grassmannian manifolds	43 subjects	Dice score = 91
Linguraru et al. [43]	Morphology-based parameterization	101 subjects	AUC = 0.85
Christ et al. [44]	CNN	100 subjects	Dice score = 94%
Han [45]	Deep CNN	200 subjects	Dice score = 67%
Vorontsov et al. [46]	Fully connected CNN	200 subjects	Dice score = 77.3%
Li et al. [47]	Hybrid CNN	221 subjects	Dice score = 82.4%
Chlebus et al. [48]	Fully connected CNN and object-based refinement	131 subjects	Accuracy = 77%
Meng et al. [49]	CNN and post-refinement	131 subjects	Dice score = 68.9%

**Table 2 cancers-14-06123-t002:** Detection and differentiation of HCC from other hepatic masses.

Study	Aim	Method	Data	Performance
Wu et al. [28]	Discrimination of HCC and HH	Radiomics-based classification	Multi-modal MRI (446 lesions)	Sensitivity = 82.2% and Specificity = 71.4%
Yasaka et al. [35]	Differentiation of typical HCC from indeterminate hepatic lesions, HH, and cysts	CNN	CE-CT (560 Subject)	Accuracy = 84%
Nie et al. [54]	Discrimination of HCC and FHN	Logistic regression analysis for radiomics-based features and clinical markers	CE-CT (119 subjects)	AUC = 0.92%
Nie et al. [55]	Discrimination of HCC and HA	Logistic regression analysis for radiomics-based features and clinical markers	CE-CT (119 subjects)	AUC = 0.94
Mokrane et al. [56]	Detection of HCC	Imaging features-based classification	CE-CT (178 subjects)	AUC = 0.66
Ponnoprat et al. [57]	Discrimination of HCC and CC	Histogram-based classification	CE-CT (257 subjects)	Accuracy = 88%
Liu et al. [58]	Detection of combined HCC-CC	Radiomics-based classification	CT and MRI (86 lesions)	AUC = 0.77
Shi et al. [59]	Detection of HCC	3 CDNs	CE-CT (342 subjects)	Accuracy = 85.6%
Cap et al. [60]	Differentiation of different types of hepatic lesions	CDN	CE-CT (517 subjects)	Accuracy = 81.3%
Hamm et al. [61]	Differentiation of different types of hepatic lesions	CNN	CE-MRI (494 subjects)	Accuracy = 91.9%
Wang et al. [62]	Differentiation of different types of hepatic lesions	CNN-based interpretation and feature maps	CE-MRI (494 subjects)	Accuracy = 88%
Zhen et al. [63]	Differentiation of different types of hepatic lesions	CNN	MRI (1411 subjects)	AUC = 0.98
Jian et al. [64]	Detection of HCC	Transfer-learning	CE-MRI (150 subjects)	Accuracy = 77%
Sun et al. [65]	Detection of small HCC lesions	Radiomics-based classification	CE-MRI (124 subjects)	Accuracy = 90.40%

**Table 3 cancers-14-06123-t003:** Prediction of microvascular invasion.

Study	Method	Data	Performance
Xu et al. [67]	Radiomics-based features	CE-CT (619 subjects)	AUC = 0.889
Bakr et al. [68]	Morphological and textural features	CE-CT (28 subjects)	AUC = 0.76
Peng et al. [69]	Radiomics-based features and clinical biomarkers	CE-CT (304 subjects)	Qualitative evaluation
Zheng et al. [70]	Morphological features and clinical biomarkers	CT (120 subjects)	AUC = 0.88
Cucchetti et al. [71]	Clinical biomarkers	Radiology and histopathology (250 subjects)	Accuracy = 88%
Jiang et al. [72]	Radiomics-based features and CNN	CE-CT (405 subjects)	Accuracy = 85.2% and AUC = 0.906
Ni et al. [73]	Radiomics-based features	CE-CT (206 subjects)	Accuracy = 84.4%
Zhang et al. [74]	Statistical analysis	CT or MRI (4759 subjects)	AUC = 0.86
Zhou et al. [75]	CNN	CE-MRI (114 subjects)	AUC = 0.926
Wang et al. [76]	Histogram-based features and clinical biomarkers	CE-MRI (113 subjects)	AUC = 0.798
Dai et al. [77]	Radiomics-based features	CE-MRI (69 subjects)	AUC = 0.895
Wang et al. [78]	CNN	DW-MRI (97 subjects)	AUC = 0.79

**Table 4 cancers-14-06123-t004:** HCC grade and molecular signature.

Study	Aim	Method	Data	Performance
Mao et al. [84]	Discrimination of high/low grade HCC	Radiomics-based features and clinical biomarkers	CE-CT (297 subjects)	AUC = 0.8014
Wu et al. [85]	Discrimination of high/low grade HCC	Radiomics-based features and clinical biomarkers	Multi-modal MRI (170 subjects)	AUC = 0.8
Zhou et al. [87]	Discrimination of high/low grade HCC	CNN	DW-MRI (98 subjects)	Accuracy = 80%
Huang et al. [90]	Detection of DPHCC	Radiomics-based classification	CE-MRI (100 subjects)	Accuracy = 79.8%
Geng et al. [91]	Differentiation of four histopathology grades	Radiomics-based classification	SWI (53 subjects)	AUC = 0.8255
Yang et al. [92]	Detection of CK-19 HCC	Radiomics-based classification	CE-MRI (257 subjects)	AUC = 0.758
Wang et al. [93]	Detection of CK-19 HCC	Radiomics-based classification	CE-MRI (227 subjects)	AUC = 0.822
Fan et al. [94]	Prediction of K-67	Radiomics-based features	Multi-modal MRI (151 subjects)	AUC = 0.922

**Table 5 cancers-14-06123-t005:** Prediction of response to ablation therapy (MWA and RFA).

Study	Aim	Method	Data	Performance
An et al. [101]	Prediction of MWA response	CNN	CE-MRI (141 subjects)	Qualitative evaluation
Hu et al. [102]	Prediction of MWA and RFA response	Radiomics-based features and clinical biomarkers	CE-MRI (160 subjects)	AUC = 0.835
Liang et al. [103]	Prediction of RFA response	Clinical biomarkers	CT (83 subjects)	AUC = 0.69
Liu et al. [104]	Prediction of RFA response	Radiomics-based deep learning	CE-US (419 subjects)	Qualitative evaluation

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
