# Peer review of "The Role of Radiomics and AI Technologies in the Segmentation, Detection, and Management of Hepatocellular Carcinoma"

_cancers, 2022, doi:10.3390/cancers14246123_

Round 1

Reviewer 1 Report

This is an excellent review paper on the role of radiomics and AI-technologies in the segmentation,detection, and management of hepatocellular carcinoma. It is well-written and structured. The cited literature is uptodate. The authors need to be congratulated on their thorough review. SOme minor points:

- A Figure on the Radiomics section could be provide illustrting the steps of the process.

- I would propose removing the paragraph "In the following section, we provide a background on different techniques of AI and.." from the introduction section. It is womewhat unecessary. 

- I would revise the relevant section to title "Limitations and future perspectives".

Author Response

We sincerely appreciate the valuable comments, suggestions, and feedback provided by the reviewer on our manuscript. We thank you, in advance, for your careful consideration of our point-by-point responses given below. The revised version has been updated carefully to address all concerns raised by the reviewer. Our point-wise response is provided below and is also reflected in the manuscript with yellow highlights for your convenience.

Reviewer #1’s Comments:

  • A Figure on the Radiomics section could be provide illustrating the steps of the process

Response:

Thanks for this valuable suggestion. Accordingly, a new figure (Figure 2) is placed at the end of the Radiomics subsection (“2.2. Radiomics”, page 7) to illustrate the two types (Deep learning-based and Feature-based) and the steps of Radiomics.

  • I would propose removing the paragraph "In the following section, we provide a background on different techniques of AI and.." from the introduction section. It is somewhat unnecessary.

Response:

Many thanks for this valuable comment. We agree with the reviewer. In the revised version, we replaced this paragraph with a structured paragraph (Section 1, page 2, lines 61-66) that summarizes the structure of the paper and the role of each section. This will gives a general idea about what will be discussed and detailed later in the article.

  • I would revise the relevant section to the title "Limitations and future perspectives".

Response:

Thanks for pointing this out. In line with the respectful reviewer, the title term had been changed to “6. Limitations and future perspectives, line 646”. Moreover, the future perspectives of AI in the field of HCC had been revised and highlighted in the revised manuscript (Page 18, Lines: 662-667).

Reviewer 2 Report

Fahmy et al., discussed the role of Artificial Intelligence (AI) in the management of hepatocellular carcinoma. The article, "The role of radiomics and AI-technologies in the segmentation, detection, and management of hepatocellular carcinoma" is very well written and appropriately summarized the recent advancements in the field. However, I have a few minor comments or suggestions for the authors:

- In page 2, line 51: the authors mentioned prestigious journals and conferences. I feel its too vague of a term for the search criteria. It would be great if authors clearly state the search criteria and what they exactly mean by "prestigious" journals (impact factor?) and conferences. 

- Authors have provided an extensive overview of AI in general for the majority of the manuscript. It is of course useful for readers not familiar with AI. If there is an issue with space- would consider trimming it.

- Authors have done an excellent job summarizing numerous studies that evaluated the role of AI in HCC- however, I would suggest adding a critical appraisal of these studies and the authors' take on the studies, which I believe is the primary purpose of the review. 

Author Response

We sincerely appreciate the valuable comments, suggestions, and feedback provided by the reviewer on our manuscript. We thank you, in advance, for your careful consideration of our point-by-point responses given below. The revised version has been updated carefully to address all concerns raised by the reviewer. Our point-wise response is provided below and is also reflected in the manuscript with yellow highlights for your convenience.

  • In page 2, line 51: the authors mentioned prestigious journals and conferences. I feel it's too vague of a term for the search criteria. It would be great if authors clearly state the search criteria and what they exactly mean by "prestigious" journals (impact factor?) and conferences.

Response:

We totally understand the concern raised by the respectful reviewer. It is worth mentioning that by “prestigious” we mean the admired/known journals and conferences in the field excluding the predatory ones. Moreover, to make this clear with the search criterion, the term “prestigious” had been removed in the revised manuscript. The updated research criterion is on the introduction section, page 2, lines 47-56.

  • Authors have provided an extensive overview of AI in general for the majority of the manuscript. It is of course useful for readers not familiar with AI. If there is an issue with space- would consider trimming it.

Response:

Many thanks for these kind encouraging words. We totally agree with the reviewer. However, one of the aims of this article is to help physicians and radiologists, a large portion of Cancers journal readers, to get familiar with the concepts of AI and ML in the field of HCC. So, the authors think that this general overview of AI, ML, and Radiomics might be useful for a targeted category of readers who are not familiar with these engineering aspects. However, the authors are welcome to trim these parts, if there is any issue with space.

  • Authors have done an excellent job summarizing numerous studies that evaluated the role of AI in HCC- however, I would suggest adding a critical appraisal of these studies and the authors' take on the studies, which I believe is the primary purpose of the review.

Response:

Many thanks to the respectful reviewer for this valuable suggestion. This comment potentially improves the quality of the revised version. Following the reviewer's suggestion, three paragraphs had been added to the revised manuscript (in addition to a new paragraph in the Limitations section, section 6) to highlight the take on the included studies concerned with different applications in the field of HCC. These added paragraphs appear in the revised manuscript in Lines: 355-361 (Section 3), 407-414 (section 4), and 641-645 (section 5). Moreover, a paragraph in the “Limitations and future perspectives” section (Lines: 662-667) had been added to highlight the overall future perspectives based on the analysis of the limitations of the included studies.

Reviewer 3 Report

This review article describes the clinical applications of radiomics and AI in the imaging diagnostics of hepatocellular carcinoma (HCC). The manuscript is well written and would be a great interest of readers of Cancers. However, there are some minor drawbacks in the manuscript.

Lines 414 and 420:

It would be useful in the authors gave specific values of accuracy.

Lines 588-601:

Recent reports found that higher enhancement intrahepatic HCC nodules on the hepatobiliary phase of Gd-EOB-DTPA-enhanced MRI could reflect the activation of the  Wnt/β-catenin signaling pathway. Furthermore, it was reported that the enhancement values could be predictors of response to ICI treatments. The authors should cite appropriate papers.

Sample literatures:

https://pubmed.ncbi.nlm.nih.gov/33083276/

https://pubmed.ncbi.nlm.nih.gov/35053606/

https://pubmed.ncbi.nlm.nih.gov/34950184/

Line 612: Please change ‘Hammaoto’ into ‘Hamamoto’.

Author Response

We sincerely appreciate the valuable comments, suggestions, and feedback provided by the reviewer on our manuscript. We thank you, in advance, for your careful consideration of our point-by-point responses given below. The revised version has been updated carefully to address all concerns raised by the reviewer. Our point-wise response is provided below and is also reflected in the manuscript with yellow highlights for your convenience.

  • Lines 414 and 420: It would be useful in the authors gave specific values of accuracy.

Response:

Thanks for this valuable suggestion. Accordingly, the reported accuracies have been added in the revised manuscript (lines 428 and lines 433-434).

  • Lines 588-601: Recent reports found that higher enhancement intrahepatic HCC nodules on the hepatobiliary phase of Gd-EOB-DTPA-enhanced MRI could reflect the activation of the Wnt/β-catenin signaling pathway. Furthermore, it was reported that the enhancement values could be predictors of response to ICI treatments. The authors should cite appropriate papers.

Sample literatures:

https://pubmed.ncbi.nlm.nih.gov/33083276/

https://pubmed.ncbi.nlm.nih.gov/35053606/

https://pubmed.ncbi.nlm.nih.gov/34950184/

Response:

We totally agree with the respectful reviewer regarding this point. Hence, this discussion with the appropriate references (Ref. 154, Ref. 155, and Ref. 156, in the revised manuscript) had been added to the end of subsection “5.4. Prediction of response to systemic treatment and/or immunotherapy” (Lines: 616-620).

  • Line 612: Please change ‘Hammaoto’ into ‘Hamamoto’.

Response:

Many thanks for pointing this out. This typo has been corrected in the revised manuscript (Line: 631).